# *Salmonella* Shedding in Slaughter Pigs and the Use of Esterified Formic Acid in the Drinking Water as a Potential Abattoir-Based Mitigation Measure

**DOI:** 10.3390/ani12131620

**Published:** 2022-06-23

**Authors:** María Bernad-Roche, Alejandro Casanova-Higes, Clara María Marín-Alcalá, Raúl Carlos Mainar-Jaime

**Affiliations:** 1Departamento de Patología Animal, Facultad de Veterinaria, Instituto Agroalimentario de Aragón-IA2, Universidad de Zaragoza-CITA, 50013 Zaragoza, Spain; mbernadroche@gmail.com (M.B.-R.); acasanova@unizar.es (A.C.-H.); 2Departamento de Ciencia Animal, Centro de Investigación y Tecnología Agroalimentaria de Aragón, Instituto Agroalimentario de Aragón-IA2, Universidad de Zaragoza-CITA, 50059 Zaragoza, Spain; cmarin@unizar.es

**Keywords:** *Salmonella*, shedding, swine, abattoir, formic acid, drinking water

## Abstract

**Simple Summary:**

*Salmonella* excretion at slaughter is considered a source of carcass contamination and human infections. To assess this potential risk, a survey on *Salmonella* shedding at slaughter in 1068 pigs from 24 farms was carried out. Almost one-third of these farms (27.3%) shed *Salmonella*. The monophasic variant of *Salmonella* Typhimurium, an emerging serotype of zoonotic importance, was the most frequent (46.9%). Antimicrobial resistance in *Salmonella* isolates was common, but resistance against antimicrobials of critical importance for humans was low, with the exception of tigecycline, a new tetracycline-derivative antimicrobial used to treat severe infections caused by extensively drug-resistant bacteria that is not used in food-producing animals. An abattoir-based strategy for the control of *Salmonella* shedding, consisting of the addition of formic acid esterified in the form of glycerides in drinking water while waiting for slaughter, was able to significantly reduce the proportion of pigs shedding *Salmonella*. It appears this strategy can contribute to mitigating the burden of abattoir environmental contamination.

**Abstract:**

Pigs shedding *Salmonella* at slaughter are considered a source of carcass contamination and human infection. To assess this potential risk, the proportion of *Salmonella* shedders that arrive for slaughter was evaluated in a population of 1068 pigs from 24 farms. Shedding was present in 27.3% of the pigs, and the monophasic variant of *Salmonella* Typhimurium, an emerging zoonotic serotype, was the most prevalent (46.9%). Antimicrobial resistance (AMR) in *Salmonella* isolates was common, but few isolates showed AMR to antimicrobials of critical importance for humans such as third-generation cephalosporins (5%), colistin (0%), or carbapenems (0%). However, AMR to tigecycline was moderately high (15%). The efficacy of an esterified formic acid in the lairage drinking water (3 kg formic acid/1000 L) was also assessed as a potential abattoir-based strategy to reduce *Salmonella* shedding. It was able to reduce the proportion of shedders (60.7% in the control group (CG) vs. 44.3% in the treatment group (TG); *p* < 0.01). After considering clustering and confounding factors, the odds of shedding *Salmonella* in the CG were 2.75 (95% CI = 1.80–4.21) times higher than those of the TG, suggesting a potential efficacy of reduction in shedding as high as 63.6%. This strategy may contribute to mitigating the burden of abattoir environmental contamination.

## 1. Introduction

Despite the efforts carried out by public health and veterinary authorities throughout Europe, salmonellosis continues to be one of the most important foodborne zoonotic diseases in the European Union (EU). In 2020, the three most-reported *Salmonella* serovars in human cases in the EU were *S*. Enteritidis, *S*. Typhimurium, and the monophasic variant of *S*. Typhimurium, the latter two being significantly related to pig sources [1]. In contrast to poultry and egg production, few European countries have established national control programs against salmonellosis in pig production, and those who tried have failed in improving the overall *Salmonella* status in their pig herds [2].

*Salmonella* is widespread among pigs [3]. An average of 33.3% of herds tested positive for *Salmonella* in 2008 in the EU according to a baseline study [4]. Since the bacteriological analysis of pig fecal samples lacks sensitivity, these results may be underestimating the true herd prevalence of infection [5]. Thus, in a context where a high proportion of pig herds are infected with *Salmonella*, it is expected that a large number of pigs will arrive to the abattoir already infected, as shown in previous surveys [4,6,7].

Although most *Salmonella*-infected fattening pigs remain asymptomatic while at farms, stressful situations such as the transport to the abattoir or the time at lairage can increase the load of *Salmonella* in feces in those previously infected pigs [8], and also increase the susceptibility to infection of the healthy ones [9]. There is evidence that on-farm *Salmonella* status has a significant influence on *Salmonella* shedding at slaughter [10,11].

Despite this expected high prevalence of *Salmonella* infection in slaughter pigs, no studies have focused on the proportion of pigs shedding *Salmonella* when they arrive for slaughter, although some have assessed the presence of the bacterium in cecal content. A study in Northern Ireland in 2002 identified *Salmonella* spp. in 31.4% of cecal content samples analyzed [12]. More recent studies carried out in the United Kingdom in 2013 and 2019 reported the presence of *Salmonella* in 30.5% and 32% of this type of sample, respectively [13,14]. The presence of *Salmonella* in cecal content would favor the presence of *Salmonella* in the distal part of the intestinal tract (colon), that is, it would favor its shedding, and further high levels of environmental contamination in the lairage area, both increasing the likelihood of carcass contamination [15,16]. Indeed, a positive correlation between high *Salmonella* cecal loads and carcass contamination has been observed [17]. Thus, to keep the proportion of *Salmonella*-contaminated carcasses low, strict slaughter hygiene practices, i.e., thorough cleaning and disinfection of facilities along with careful eviscerations, need to be intensified. In addition, activities aimed at reducing the *Salmonella* burden in the guts (shedding) of fattening pigs once they reach the abattoir should also help to further reduce the proportion of contaminated carcasses.

After transport, pigs usually arrive at the abattoir thirsty, creating the need for drinking water. This condition may be an opportunity for enhancing the ingestion of water treated with esterified organic acids (OAs) before entering the slaughter line. Short and medium chain fatty acids combined with glycerol have enhanced antimicrobial activity against Gram-negative bacteria both in vitro and in vivo [18]. In addition, the esterification of these acids has shown additional advantages as they are less pH-dependent and less susceptible to enzymatic breakdown, allowing them to act along the whole gastrointestinal tract [19]. In addition, due to its glyceride form, the OAs used in this way are odorless and non-corrosive, and their amphipathic structure make them soluble in water without changing their pH, thus avoiding any off-flavors in the water that would prevent the pigs from drinking it. Therefore, they may be of help in reducing the gastrointestinal burden of Gram-negative bacteria such as *Salmonella*, provided the proper dosage is given and enough time is considered for the OAs to take effect.

Thus, two studies with two main objectives were designed. In the first study, the prevalence of *Salmonella* shedding, namely, the proportion of pigs showing *Salmonella* in their feces at slaughter, in a population of pig herds from a region considered to have a high prevalence of pig salmonellosis, was estimated. Furthermore, the patterns of antimicrobial resistance (AMR) against antibiotics of importance for humans of the *Salmonella* strains isolated from these pigs were described. Thus, a picture of the risk potentially faced by abattoirs could be obtained. In the second study, it was assessed whether the addition of an esterified formic acid to the abattoir drinking water during lairage may be useful to reduce the proportion of pigs shedding *Salmonella*, so that it can be included as an additional abattoir-based *Salmonella* mitigation strategy.

## 2. Materials and Methods

### 2.1. Study 1 Prevalence Survey and Salmonella Characterization

#### 2.1.1. Herd and Animal Selection and Sample Collection

Between December 2019 and November 2021, a convenience sampling of 24 fattening pig units (1000–2000 pigs/unit) was carried out in the Northeast region of Spain. Farms were selected based on the farmer’s willingness to collaborate in a larger project to assess new strategies for the control of pig salmonellosis. From each farm, a total of 50 animals from the first batch of pigs to be sent to slaughter (the heaviest pigs) from a given fattening unit were ear tagged 3–4 weeks before leaving the farm. They were selected from different pens among the fattening units. Once in the abattoir, they were identified in the slaughter line and, after evisceration, the gastrointestinal tract was individually collected and a minimum of 25 g of colon content was taken from each of these pigs for *Salmonella* isolation. Samples were then transported to the laboratory for immediate processing.

Additional information on the biosecurity level of each of these farms was also available as it had been previously obtained through a questionnaire (Biocheck.UGent scoring system; https://biocheckgent.com/sites/default/files/2020-02/Pigs_EN_1.pdf (accessed on 6 June 2022)) that had been filled in by the responsible veterinarians of the farms. Thus, a standardized summary measure of farm biosecurity, that is, “Total biosecurity score”, was available for comparison purposes. Detailed information regarding this summary measure can be found elsewhere (https://biocheckgent.com/en/features (accessed on 6 June 2022)).

#### 2.1.2. Salmonella Isolation, Serotyping and Antimicrobial Susceptibility Testing

Intestinal content (IC) samples obtained from the distal colon of individual pigs at the evisceration line were processed and cultured for *Salmonella* isolation following the standard ISO 6579:2002/Amd 1:2007 method. Briefly, 25 g of IC colon was mixed with 225 mL of buffered peptone water (BPW ISO, Oxoid Ltd., Basingstoke, Hants, UK) and incubated at 37 ± 2 °C for 18 ± 2 h for non-selective pre-enrichment. Following incubation, 100 µL divided into three drops of approximately 33 µL was spotted onto a Modified Semisolid Rappaport-Vassiliadis agar (MSRV, Oxoid Ltd., Basingstoke, Hants, UK) plate and incubated at 41.5 ± 1 °C for 24–48 h for selective enrichment. If migration zones were present on the MSRV plates, a loopful of the edge migration zones was transferred onto xylose lysine desoxycholate (X.L.D. medium, Oxoid Ltd., Basingstoke, Hants, UK) and brilliant green (BGA modified, Oxoid Ltd., Basingstoke, Hants, UK) agar plates for selective culture. Suspected colonies were confirmed phenotypically based on the biochemical reactions using tryptone soya broth (Oxoid Ltd., Basingstoke, Hants, UK), lysine descarboxylase broth (BD DifcoTM, Detroit, MI, USA), urea broth (BD DifcoTM, Detroit, MI, USA), and triple sugar iron agar (Sigma-Aldrich, St. Louis, MO, USA).

A colony from each *Salmonella*-positive culture was then selected for identification of the two major serotypes of concern in the pig industry, namely, *S*. Typhimurium and the monophasic variant of *S*. Typhimurium. For that purpose, a duplex PCR that allows simultaneous amplification of a fragment between the genes *fljB* and *fljA* and the phase-2 flagellar gene (*fljB*) was used [20,21]. The PCR was performed in 1X PCR buffer Taq, 2.5 mM MgCl_2_, 0.6 mM dNTPs, and 1 U Taq Biotools DNA polymerase (final volume of 25 µL). The primers mix contained primers at a concentration of 0.1 µM for FFLIB (5′-CTGGCGACGATCTGTCGATG-3′) and RFLIA (5′-GCGGTATACAGTGAATTCAC-3′) and at a concentration of 1.0 µM for sense-59 (5′-CAACAACAACCTGCAGCGTGTGCG) and antisense-83 (5′-GCCATATTTCAGCCTCTCGCCCG-3′). The cycling parameters involved denaturation at 95 °C for 2 min, followed by 30 cycles of 95 °C for 30 s, 64 °C for 30 s and 72 °C for 1.5 min for amplification, and a final step of 72 °C for 10 min. PCR products were separated on 2% (*w/v*) agarose gels, stained with GelGreen and visualized using a gel image system (iBright 1500, Invitrogen, Termofischer, Singapore). Thus, each *Salmonella*-positive sample was classified as *S*. Typhimurium, the monophasic variant of *S*. Typhimurium, or *Salmonella* serotypes different from these two (“other”).

A selection of at least one strain of each *Salmonella* serotype isolated from each sampled pig herd was tested for antimicrobial resistance (AMR) to further characterize them. Antimicrobial agents were chosen among those considered of critical importance for humans following the Commission Implementing Decision (EU) 2020/1729. The minimum inhibitory concentration (MIC) to penicillins (A; ampicillin, amoxicillin-clavulanic acid, piperacillin-tazobactam), aminoglycosides (S; amikacin, gentamicin), sulphonamides and dihydrofolate reductase inhibitors (Su; trimethoprim-sulfamethoxazole), tetracyclines (T; tigecycline), cephalosporins (Cf; 2nd generation: cefuroxime, cefoxitin; 3rd generation: cefotaxime, ceftazidime; 4th generation: cefepime), carbapenems (Cm; ertapenem, imipenem), and quinolones (Na; nalidixic acid, ciprofloxacin) was assessed by the VITEK-2 automated system with the VITEK AST-N243 cards (BioMérieux, Marcy-l’Étoile, France). *Salmonella* strains were further classified as resistant or susceptible according to the recommendations of the European Committee on Antimicrobial Susceptibility Testing [22]. Colistin resistance was also assessed and, in this case, MICs were determined by the broth microdilution method according to the ISO 20776-1:2006 standard. An epidemiological cutoff (ECOFF) value of >2 mg/L was used for considering microbiological resistance according to the recommendations of EUCAST. *Escherichia coli* ATCC 25922 was used as quality control strain. An isolate displaying phenotypic resistance to at least three antimicrobial classes was considered multidrug resistant (MDR) [23].

### 2.2. Study 2 Assessment of the Efficacy of Esterified Formic Acid in Abattoir Drinking Water

#### 2.2.1. Farm and Animal Selection and Sample Collection

Pig farms that had been identified as *Salmonella*-seropositive through previous routine analyses by the abattoir were chosen for this study. All of them were in the vicinity of the abattoir (average distance of 33.5 km). From each farm, pigs arrived at the abattoir in batches of approximately 200 animals (i.e., the truck load capacity). As they were unloaded, 40 were allocated to a clean lairage pen where a four-trough drinking system connected to a 1000 L water container had been set up for the treated water (treatment group, TG). The remainder of the pigs from the truck were allocated to 3–4 clean regular pens with untreated drinking water (control group, CG). Within a batch, all the pigs were of the same age and similar weight, and had been housed within the same fattening unit. In addition, the cleaning of the lairage pens was carried out in a similar way for all of them. All the pigs arrived at the abattoir during the evening of a given day and remained there until slaughter the next morning, a median time of 14.5 h (minimum 10 h, maximum 17.5 h) of lairage. A proxy of the total water consumed by each TG was estimated from the water left in the container after pigs left the pen.

Collection of samples from the 40 pigs from the TG and from another 40 from the CG was performed after evisceration, as indicated in Section 2.1.1. This number of pigs per group was a trade-off between our research team´s capacity to process the samples on the same day of collection and the statistical power to detect a difference of 25 percentage points between the proportion of shedders in the control group (estimate of shedding of 30–35% according to previous studies on infection prevalence) and the treatment group.

Trials were carried out in spring time (3), summer (1), and autumn (3), depending on abattoir availability.

#### 2.2.2. Organic Acids

A formic acid (30% formic acid) esterified in the form of mono-, di- and triglyceride with glycerol (MOLI-M C1, Molimen SL, Barcelona, Spain) was selected for this study. Mono-, di- and triglycerides of fatty acids are within the catalogue of feed materials approved by the European Commission (Commission Regulation (EU) No 68/2013 of 16 January 2013 on the Catalogue of feed materials).

Given the expected short period of time during which pigs would be exposed to the treated water, a higher inclusion dose (10 kg product/1000 L of water) than that usually used at the farm (3.5 kg product/1000 L of water) was considered for this study. This dose was still much below the maximum value of 12 g/kg formic acid approved for the use in pigs in the European Union (EU) [24].

#### 2.2.3. *Salmonella* Isolation

Isolation of *Salmonella* from IC samples was performed as indicated above. No further characterization was carried out.

### 2.3. Statistical Analyses

For the survey, estimates of prevalence of *Salmonella* shedding at slaughter with its corresponding 95% confidence intervals (95% CI) were calculated overall and for each individual pig herd. Differences among seasons were also assessed.

Overall levels of antimicrobial resistance among the three groups of serotypes considered (*S*. Typhimurium, the monophasic variant of *S*. Typhimurium, and “other”) were compared using the summary measure for antimicrobial resistance used by Poppe et al. [25]. This measure was expressed as the percentage of the average resistance to the 18 antimicrobial agents to which resistance of the isolates was tested, and was calculated as:$$\mathrm{PR}=\frac{\left(\mathrm{TotResist}\right)/17}{\mathrm{TotIsolates}}$$
where PR means percent resistance; TotResist is the sum of all measures of resistance for a given set of isolates to any of the 18 antimicrobial agents in the test panel; and TotIsolates is the total number of isolates examined. In addition, the proportion of MDR isolates within each serotype group were estimated and compared by chi-squared test.

With regard to the assessment of the efficacy of the esterified formic acid, in a first step, for each trial, the prevalence of *Salmonella* shedding was compared between the CG and the TG by the Fisher’s exact test. Afterward, a random-effects logistic regression analysis considering results from all the trials was performed to obtain an overall estimate of the effect of the treatment on the prevalence of *Salmonella* shedding. In this analysis the presence of *Salmonella* in the colon content was the dependent variable and “Treatment” the independent variable. “Trial” was considered as a random variable to account for correlation between individuals within the same trial (i.e., intra class correlation, ICC), and the qualitative variables “Season” (spring, summer or autumn) and “Time spent at lairage” (with two categories based on the median value; <15 h vs. ≥15 h) were considered as potential confounding factors and forced into the model. The odds of *Salmonella* shedding in the CG were compared to those in the TG through estimation of the Odds Ratio (OR) and its corresponding 95% CI.

From this adjusted OR, an estimate of the attributable fraction, that is, the proportion of shedding pigs in the CG that was due to the absence of treatment or, that is, the proportion of pigs that may have been prevented from shedding through the use of this OA during the lairage, could be estimated as OR-1/OR [26].

In addition, to assess whether the estimated water consumption per pig in the TG was associated with the proportion of *Salmonella* shedders, the mean water consumption per pig in each TG was calculated from the water left in the container. Further, the median of the mean water consumption per pig was obtained and the proportions of *Salmonella* shedders among pigs drinking below and above the median were compared by a chi-squared test. OR and its 95% confidence interval were also estimated.

The software STATA (STATA/IC 12.1. Stata-Corp. LP, College Station, TX, USA) was used for statistical analyses.

## 3. Results

### 3.1. Study 1 Prevalence Survey and Salmonella Characterization

#### 3.1.1. Prevalence of Salmonella Shedding

A total of 1068 pigs of the 1200 initially ear tagged at the farm were identified at the slaughterhouse. Therefore, an average 44.5 pigs were analyzed per herd for the presence of *Salmonella* (Table 1). The pigs were slaughtered in two different large-scale abattoirs (292 in slaughter A and 776 in B).

Twenty-two farms (91.7%; 22 of 24) had at least one *Salmonella*-positive pig. *Salmonella* was recovered from 292 pigs (27.3%; 95% CI = 24.8–30.1). The major serotype identified was the monophasic variant of *S*. Typhimurium, which was isolated in fecal samples from 137 pigs (46.9%; 95% CI = 41.3–52.6) and was detected in 54.5% (12/22) of the pig herds. *Salmonella* Typhimurium was identified only in 55 pigs (18.8%; 95% CI = 14.8–23.7), but well distributed among herds (50%). One hundred isolates belonged to “other” serotypes and were present in 72.7% of the herds. Only seven herds were infected with serotypes other than *S*. Typhimurium or its monophasic variant. Overall, the mean *Salmonella* prevalence in these positive herds was 29.8%. *Salmonella* prevalence was significantly different among seasons (*p* < 0.001), with 44.2% of the positive samples in winter, 28.4% in spring, 14.3% in summer, and 19.8% in autumn.

#### 3.1.2. Antimicrobial Susceptibility Testing

A selection of 80 isolates (20 of *S*. Typhimurium, 28 of the monophasic variant of *S*. Typhimurium, and 32 of “other” serotypes) representing the different serotypes detected in each farm were tested for antimicrobial resistance (AMR) against the 17 antimicrobial agents described above (Table 2).

Resistance to ampicillin was the most common (73.8%) AMR observed in *Salmonella* isolates, varying from very high levels (59.4%) in “other” serotypes to extremely high (70–92.9%) in *S*. Typhimurium and its monophasic variant. Resistance to trimethoprim-sulfamethoxazole and nalidixic acid was observed at moderate levels (≈20%) but was well distributed among herds (36.4% and 45.4%, respectively) and involving all serotypes. Much lower resistance was detected against the aminoglycosides, with no resistance observed against amikacin and only 6.2% of the isolates were resistant to gentamicin.

Resistance to tigecycline was present in 15.0% of the isolates, and distributed among all serotypes. However, resistance was low against cephalosporins of the third generation (5% for cefotaxime, and 3.8% for ceftazidime) and ciprofloxacin (1.2%), and was null against carbapenems (imipenem and ertapenem) or polymyxins (colistin). Although resistance to tigecycline was present in five (22.7%) of the herds, resistance to third-generation cephalosporins was detected in only two (9%) of them, and mostly affected “other” serotypes.

Overall, MDR was observed in 11 (13.8%) of the isolates. Among these isolates, 54.5% were “other” serotypes, 27.3% belonged to the monophasic variant of *S*. Typhimurium and 18.2% to *S*. Typhimurium, although no significant differences were observed between the proportion of MDR isolates among group of serotypes (*p* = 0.56).

In five (45.4%) of these MDR isolates, resistance to tigecycline was present. No combined resistance to both ciprofloxacin and cefotaxime was observed.

Complete susceptibility was detected in 25% of *Salmonella* isolates with the highest proportion in “other” serotypes (37.5%), followed by *S*. Typhimurium (30.0%). Only two (7.1%) isolates of monophasic variant of *S*. Typhimurium were pansusceptible.

Regarding AMR profiles, 12 different patterns were detected, the most common being resistance against aminopenicillins (*n* = 24; 30%), followed by aminopenicillins-quinolones (*n* = 8; 10%) and aminopenicillins-sulfonamides (*n* = 6; 7.5%). ASSuNa, ATCf, ASuTNa, ASSuCf, and ASuNa profiles were detected in 5%, 3.8%, 2.5%, 1.3% and 1.3% of the isolates, respectively.

Percent resistance among isolates of the monophasic variant of *S*. *typhimurium* was 10.7%, somewhat higher than that for *S*. Typhimurium (6.8%) but similar to that of “other” serotypes (10.3%).

### 3.2. Study 2 Assessment of the Efficacy of Esterified Formic Acid in Abattoir Drinking Water

A total of seven trials (on seven different pig herds) were performed. Mean prevalence of *Salmonella* shedding was variable among trials, ranging from 18.8% in trial 3 to 90% in trial 2. Results for each group in each trial are shown in Figure 1. In general, *Salmonella* shedding levels were always lower in the TG compared to the CG and, at the trial level, in three of them these differences were statistically significant.

The proportion of *Salmonella* shedders in those pigs that consumed an amount of treated water below the median (0.9 L/pig) was significantly higher than that when water consumption was above the median. The odds of shedding *Salmonella* in those pigs was up to 4.56 times higher than that in the group that drank ≥0.9 L of treated water (Table 3).

Overall, in 60.7% (170/280) of the pigs from the CG, *Salmonella* was found in feces. This percentage decreased to 44.3% (125/282) in the TG (*p* < 0.01). The random-effects logistic regression analysis indicated a significant clustering effect of “Trial” (ICC: 0.302; *p* < 0.001). According to the model, *Salmonella* shedding significantly decreased with the treatment after adjusting by season and lairage time (Table 4). The adjusted odds of shedding *Salmonella* in the CG were 2.75 (95% CI = 1.80–4.21) times higher than those in the TG. Thus, the estimated proportion of shedders that may have been prevented due to the use of this esterified formic acid in this population may be as high as 63.6%.

## 4. Discussion

Pig salmonellosis is mostly a public health issue as pigs may become a major source of infection for humans [1]. Although its transmission by direct contact with infected pigs is possible, the main route of transmission is through the consumption of *Salmonella*-contaminated pork products. Contamination of pork products usually occurs during the processing of the carcass at slaughter. According to the relevant authorities, the proportion of contaminated pig carcasses in Spain in 2019 and 2020, a country with some of the highest levels of pig salmonellosis, was as high as 17.6% and 14.3%, respectively [1,27]. Trying to get a better insight into the reasons why this is occurring thus appears necessary. In addition, most European national control programs aiming at reducing the prevalence of *Salmonella* at the farm level, as a first step to reduce its prevalence at the slaughter level, have failed [2,28]. Therefore, abattoir strategies that may help to mitigate this problem must be sought while more efficient farm control programs are developed.

Previous studies have estimated the proportion of *Salmonella*-infected pigs that reach the abattoir [4,7], but no one has assessed how many of these arrive shedding the bacterium, which would directly relate to abattoir contamination. According to the results from the present study, more than 90% of the herds sent pigs that shed *Salmonella* at slaughter, and 27.3% of the slaughtered pigs showed *Salmonella* in their feces. Although some pigs may have become infected during transport or even during lairage [29], from the abattoirs´ point of view, almost one-third of the pigs arriving could be considered a potential source for carcass contamination.

Since the survey was based on a convenience sampling, these results cannot be considered representative of a given pig farm population and were likely subject to selection bias. We checked data available on the farm biosecurity level to assess whether this issue may explain the high levels of shedding found. On average, the farms included in this study presented reasonable levels of biosecurity, that is, a “Total biosecurity score” of 72% (of 100%, which would mean perfect farm biosecurity), ranging from a minimum of 51% to a maximum of 86% (data available upon request). This average was above many national averages (63% for German pig farms; 65% for Italian; 67% for Spanish; 69% for Dutch) and similar to those from Irish or Belgian farms (72% and 74%, respectively) (checked at https://biocheck.ugent.be/en/worldwide on 12 April 2022). Although much caution should be considered with such comparisons, as there was no knowledge on the origin and representativeness of the farms included in the cited web page, this comparison provides a glimpse into the ranking of our farms, and suggests that the high proportion of pigs shedding *Salmonella* observed in this study should not be related to a selection bias associated with farms of low biosecurity levels. Neither should it be attributed to the quality of the abattoirs involved, since both were new modern slaughterhouses with up-to-date robotized meat processing techniques and high levels of hygiene.

These results agree with those of a recent study carried out in another Spanish region, where 71.4% of batches of slaughter pigs (pooled fecal samples) were contaminated upon arrival [30]. In general, these results support the idea that the situation remains stable and no improvement has been observed since the last EFSA baseline study performed more than 10 years ago. Considering the expected prevalence of *Salmonella* infection in the country according to that study (≈30%; [4]), and the proportion of shedding pigs observed at slaughter in this one (27%), it appears that a relationship between field infection and shedding at abattoir can be assumed.

The most prevalent serotype detected in feces was the monophasic variant of *S*. Typhimurium, a serotype of high public health importance, and present in 47% of the pigs shedding *Salmonella*; this far exceeded not only *S*. Typhimurium (18.8%), but also the sum of the rest of the “other” serotypes (34%). This represents a significant jump in the prevalence of the monophasic variant of *S*. Typhimurium with respect to previous studies in the same or neighboring areas, when its prevalence was as low as 12% [7,31], and confirms the worldwide upward trend of this serotype [1,30,32,33,34,35,36,37].

This rise in the prevalence of the monophasic variant of *S*. Typhimurium appears to be related with some evolutionary advantage, in part likely associated with the accumulation of mobile resistance elements, which in turn would have contributed to its monophasic phenotype [38,39]. In fact, the monophasic variant of *S*. Typhimurium has been usually characterized by displaying AMR to ampicillin, streptomycin, sulphonamides, and tetracycline (ASSuT) [7,40,41,42]. Subsequently, it has also been related to resistance to critical antimicrobials for humans, such as third-generation cephalosporins, fosfomycin, and colistin [43,44,45], which has raised concerns about the limitation of antimicrobial therapeutic options. In this study, we used a summary measure for AMR, the so-called percentage resistance (PR) [25], to determine whether there were differences in overall AMR against the 17 antimicrobial agents tested among the three groups of serotypes considered. A somewhat higher PR was observed for the monophasic variant of *S*. Typhimurium isolates (10.7%) when compared to *S*. Typhimurium (6.8%), but this was similar to that for all the “other” serotypes together (10.3%). In any case, this level of AMR among the monophasic variant of *S.* Typhimurium isolates seemed not to be associated with a higher proportion of MDR isolates, nor with resistance to antimicrobials of critical importance for humans, since no isolate showed AMR against carbapenems or colistin, and just one showed it to third-generation cephalosporins.

Carbapenems are not used in animals and no resistance against this antimicrobial class has been observed in *Salmonella* isolates from pigs in the EU in recent years [46]. Resistance to colistin has been usually low in Spain, even when this drug was used massively in the pig sector [44,47]. The application of the Spanish Plan against Antibiotic Resistance in 2015, and the subsequent reduction in colistin consumption by more than 95% [48], may have contributed to an overall reduction in the detection of the *mcr*-1 gene [49,50], explaining the absence of phenotypic resistance against colistin in these isolates. Regarding AMR to third-generation cephalosporins, this was found in 5% of the *Salmonella* isolates. Although this level was still higher than the European average of 0.5% [51], it appears constant with regard to previous years [47,52], and was circumscribed to only two farms, suggesting its presence may have been triggered by the occasional use of this class of antimicrobials (e.g., ceftiofur) in these farms [52].

In general, the observed levels of resistance against the antimicrobials tested in this group of isolates was similar to that described in the last report from EFSA and ECDC regarding antimicrobial resistance in zoonotic and indicator bacteria from humans, animals, and food in 2018–19 [51], with the exception of two antibiotics, namely, ciprofloxacin and tigecycline. Resistance to ciprofloxacin, a first-line antimicrobial agent for the treatment of non-typhoidal *Salmonella* in humans, was observed at lower levels (1.2%) than those noticed in pigs in the EU (10.3%; [46]), and at much lower levels than those recently observed in Italy (60.7%; [53]). This type of resistance in *Salmonella* has been related to fluoroquinolone consumption in food-producing animals [54,55]; thus this reduction may be related to a decreasing trend in its use, as the last European Surveillance of Veterinary Antimicrobial Consumption (ESVAC) report on sales of veterinary antimicrobial agents suggests [56].

On the contrary, resistance to tigecycline, a tetracycline-derivative antibiotic used to treat severe infections caused by extensively drug-resistant bacteria [57], was moderately high (15%), well spread (in five different herds), and much higher than that shown in the European data (1.5%; [46]). Resistance to this drug has been recently associated to highly transferable plasmid-mediated genes detected in bacteria from human and animals, including pigs [58,59], which may explain its spread within the swine sector despite the fact that tigecycline is not used in veterinary medicine. Considering that five (41.7%) of the tigecycline-resistant isolates showed MDR, co-selection for resistance to this antimicrobial through exposure to other antimicrobials may also have been possible.

A farm-to-consumption quantitative microbiological risk assessment (QMRA) already suggested that a large proportion of cases of human salmonellosis would derive from slaughter pigs with a high concentration of *Salmonella* in their feces [60], which likely occurs due to the lack of efficient *Salmonella* control measures at the farm level [2]. Therefore, the implementation of control steps at the abattoir to minimize the transfer of *Salmonella* to the surface of pig carcasses appears compulsory, particularly since EU legislation forbids the treatment of carcasses with antimicrobial agents (Regulation EC 853/2004). Strict hygiene measures along with the reduction in *Salmonella* shedders are the only measures foreseen to reduce abattoir contamination, the former being more effective depending upon the reduction that can be reached on the latter.

Thus, the present study also aimed at assessing whether the addition of an esterified formic acid to the lairage drinking water can help to reduce the overall proportion of pigs shedding *Salmonella* while waiting for slaughter. The net amount of formic acid used (3 g/kg) was significantly below the maximum dose of formic acid considered safe for growing pigs (18 g/kg feed; [61]). Using this dose, a significant overall reduction in the number of shedders in the TG was observed when the seven trials were taken together (44% in the TG vs. 60.7% in the CG), with a decreasing trend in all the trials; however, only in three of them was this drop was statistically significant.

When OAs or their salts have been applied at the farm level, their efficacy for the control of pig salmonellosis has been variable. Although some studies have shown a reduction in *Salmonella* excretion [62,63,64,65], or seroprevalence [63,66,67,68,69,70,71,72], others did not find any effect at all [73,74,75]. It is well known that the antimicrobial effect of OAs depends upon factors such as the type and combination of OA, the concentration used, the time and period of administration, and the age of the animals. In general, however, long periods of treatment seem to be required before any positive effect is detected [71]. In this study, however, overnight drinking (between 10 and 17 h of exposure to the treated water) was enough to detect a significant beneficial effect. Reasons for this finding may be related to the type of formulation used, that is, formic acid esterified in the form of glycerides. Esterified acids appear to have an increased and thorough antimicrobial activity along the whole intestinal tract compared to the original OAs or their salts [18,19]. In addition, it did not modify the water taste despite the dose considered. Since the main mechanism of action of esterified fatty acids relates to a membrane-destabilizing activity that causes increased cell permeability and lysis after contact with the bacterium [76], it is also possible that its effect could begin within the oral cavity.

The lack of a significant reduction in *Salmonella* shedding in some of the trials may be related to other factors such as the amount of water consumed by the pigs or the length of the lairage time. Although it was not possible to accurately measure individual water consumption, a proxy of that was estimated based on water left in the container for each treated group. It was found that the chances for shedding *Salmonella* increased significantly when pigs in these groups had drunk less than 0.9 L of treated water (OR = 4.56; 95% CI: 2.70–7.69). Finding strategies that favor the consumption of treated water at lairage (i.e., increasing the number of drinking troughs) may help to increase the efficacy of this strategy.

Similarly, prolonged lairage times appear to favor the risk of *Salmonella* shedding and, therefore, environmental contamination [77]. In this study, all batches of pigs were kept overnight and for a minimum of 10 h to favor the consumption of water, which may also contribute to somewhat neutralizing the positive effect of the treatment applied. Considering the apparent beneficial effect of this esterified formic acid, beginning the water treatment at the farm, a few days before the slaughter, would likely avoid the need for long periods of lairage.

The product tested allowed a raw reduction of around 16% in the proportion of *Salmonella* shedding pigs, compared to a CG without the treatment. This drop may seem small, but it happened in a context of high prevalence of shedding (>60% in the CG) and after a few hours of treatment. Other strategies, such as vaccination, have shown efficacies that were only slightly higher, i.e., 20–30% [78,79]. Interestingly, when other variables were taken into account (clustering and confounding variables), the OR of shedding for the CG vs. the TG was much higher than that when no adjustment was considered (2.75 vs. 1.94). Therefore, the proportion of pigs that may have been prevented from shedding due to the use of this esterified formic acid in the TG, given the conditions of the study, may be as high as 63.6%. Thus, this strategy can potentially contribute to reduce the burden of *Salmonella* contamination in the slaughter premises and therefore be used as a complement within more comprehensive *Salmonella* control programs that include both on-farm and abattoir-based strategies.

## 5. Conclusions

A high proportion of pigs (18%) still arrives at the abattoir shedding important zoonotic *Salmonella* serotypes (mostly *S*. Typhimurium and its monophasic variant). In addition, in some cases, the salmonellae detected display AMR against antimicrobials of critical importance for humans. Although, in general, the AMR to these classes of antimicrobial was low, the levels of AMR against tigecycline, an antimicrobial not used in food-producing animals, were of concern. These shedding pigs contribute to the abattoir environmental contamination and are a further potential source for human infections. Slaughter strategies such as the use of formic acid esterified in the form of glycerides in the drinking water may help to reduce the burden of environmental contamination through the reduction in the proportion of shedding pigs prior to slaughter, but further research on the efficacy of this product is needed.

## Figures and Tables

**Figure 1 animals-12-01620-f001:**
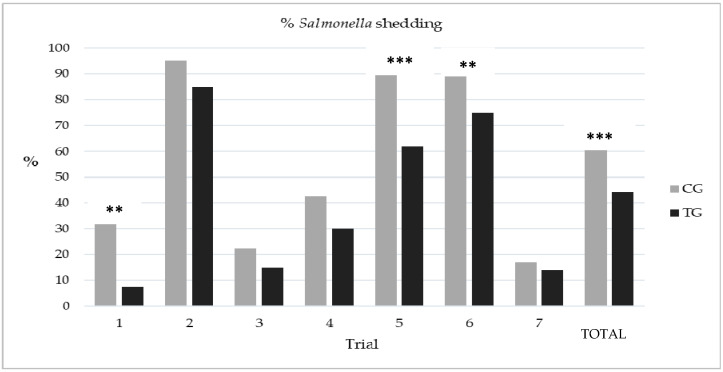
Proportion of *Salmonella* shedding at slaughter in the seven trials and globally. Significant differences between the control (CG) and the treated group (TG) are indicated (** *p* < 0.05; *** *p* < 0.01; Fisher´s exact test).

**Table 1 animals-12-01620-t001:** Prevalence of *Salmonella* shedding and major serotypes found in slaughter pigs from 24 farms.

Farm ID	N	No. + (%)	Serotypes Involved (No. of Strains)
1	43	0 (0.00)	-
2	46	19 (41.30)	*S*. Typhimurium (3), mST (16)
3	45	1 (2.22)	other (1)
4	49	9 (18.37)	*S*. Typhimurium (1), mST (8)
5	46	35 (76.09)	*S*. Typhimurium (20), mST (5), other (10)
6	48	27 (56.25)	*S*. Typhimurium (7), mST (18), other (2)
7	25	18 (72.00)	other (18)
8	48	33 (68.75)	*S*. Typhimurium (11), mST (5), other (17)
9	46	15 (32.61)	*S*. Typhimurium (1), mST (7), other (7)
10	45	39 (86.67)	*S*. Typhimurium (4), mST (34), other (1)
11	50	4 (8.00)	mST (2), other (2)
12	51	1 (1.96)	*S*. Typhimurium (1)
13	44	8 (18.18)	mST (8)
14	48	9 (18.75)	other (9)
15	48	7 (14.58)	mST (7)
16	47	5 (11.36)	other (5)
17	46	4 (8.70)	other (4)
18	39	1 (2.56)	other (1)
19	34	27 (79.41)	mST (25), other (2)
20	46	1 (2.17)	other (1)
21	44	2 (4.55)	*S*. Typhimurium (1), other (1)
22	44	19 (43.18)	*S*. Typhimurium (3), mST (2), other (14)
23	44	0 (0.00)	-
24	42	8 (19.05)	*S*. Typhimurium (3), other (5)
All	1068	292 (27.34)	*S*. Typhimurium (55), mST (137), other (100)

mST: monophasic variant of *S*. Typhimurium; “other”: serotypes other than *S.* Typhimurium and its monophasic variant.

**Table 2 animals-12-01620-t002:** Occurrence of resistance to critically important antimicrobials in *Salmonella* isolates from pigs.

Antimicrobial Class	Antimicrobial Agent *	No. ST (%) *n* = 20	No. mST (%) *n* = 28	No. “Other” Serotypes (%) *n* = 32	Farm ID
Penicillins	AMP	14 (70.0)	26 (92.9)	19 (59.4)	2, 4, 5, 6, 8, 9, 10, 11, 12, 13, 15, 16, 17, 19, 20, 21, 22, 24
	AMC	0 (0.0)	3 (10.7)	3 (9.4)	11, 13
	TZP	0 (0.0)	2 (7.1)	0 (0.0)	11
Cephalosporins	CXM	1 (5.0)	1 (3.6)	3 (9.4)	13, 22
	FOX	0 (0.0)	0 (0.0)	0 (0.0)	-
	CTX	0 (0.0)	1 (3.6)	3 (9.4)	13, 22
	CAZ	0 (0.0)	0 (0.0)	3 (9.4)	22
	FEP	0 (0.0)	0 (0.0)	0 (0.0)	-
Carbapenems	ETP	0 (0.0)	0 (0.0)	0 (0.0)	-
	IPM	0 (0.0)	0 (0.0)	0 (0.0)	-
Polymyxins	CST	0 (0.0)	0 (0.0)	0 (0.0)	-
Aminoglycosides	AMK	0 (0.0)	0 (0.0)	0 (0.0)	-
	GEN	0 (0.0)	3 (10.7)	2 (6.3)	8, 11, 13
Tetracyclines	TGC	4 (20.0)	4 (14.3)	4 (12.5)	2, 11, 16, 17, 22
Quinolones	NAL	2 (10.0)	7 (25.0)	7 (21.9)	8, 9, 11, 12, 13, 16, 19, 20, 22, 24
	CIP	0 (0.0)	0 (0.0)	1 (3.1)	16
Sulphonamides dihydrofolate reductase inhibitors	SXT	2 (10.0)	3 (10.7)	11 (34.4)	8, 9, 11, 12, 13, 16, 17, 19

ST: *Salmonella* Typhimurium; mST: monophasic variant of *S*. Typhimurium. * Ampicillin (AMP), amoxicillin-clavulanic acid (AMC), piperacillin-tazobactam (TZP), cefuroxime (CXM), cefoxitin (FOX), cefotaxime (CTX), ceftazidime (CAZ), cefepime (FEP), ertapenem (ETP), imipenem (IPM), colistin (CST), amikacin (AMK), gentamicin (GEN), tigecycline (TGC), nalidixic acid (NAL), ciprofloxacin (CIP), trimethoprim-sulfamethoxazole (SXT).

**Table 3 animals-12-01620-t003:** Number of pigs shedding *Salmonella* and estimated water consumption per pig.

		Estimated Water Consumption Per Pig		
		<0.9 L	≥0.9 L *	
		No. (%)	No. (%)	Total
*Salmonella* shedding	Yes	96 (76.8)	29 (23.2)	125
	No	66 (42.0)	91 (58.0)	157
		OR = 4.56 (95% CI: 2.70–7.69; *p* < 0.0001)		

* Categories based on the median of the estimated mean water consumption per pig.

**Table 4 animals-12-01620-t004:** Results of the random-effects logistic regression analysis to assess the relationship between water treatment with the esterified formic acid and *Salmonella* shedding after adjusting by season and time at lairage.

			Logistic Regression Parameters				
Variable	N	No. + (%)	β	SE (β)	*p*	OR	95% CI OR
Group							
Treatment ^1^	282	125 (44.3)				1	
Control	280	170 (60.7)	1.01	0.21	<0.001	2.75	1.80–4.21
Season							
Spring ^1^	240	156 (65)				1	
Summer	80	31 (38.7)	−3.32	2.06	0.107	0.036	0.0006–2.05
Autumn	242	108 (44.6)	−2.45	1.42	0.085	0.086	0.005–1.39
Lairage time							
<15 h	320	171 (53.4)					
≥15 h	242	124 (51.2)	1.94	1.50	0.195	7.01	0.36–132

^1^ Reference categories. Trial (random variable): variance 1.422; intra class correlation (ICC): 0.30 (95% CI: 0.12–0.56).

## Data Availability

Data available upon request.

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
