# Peer review of "Salmonella Shedding in Slaughter Pigs and the Use of Esterified Formic Acid in the Drinking Water as a Potential Abattoir-Based Mitigation Measure"

_animals, 2022, doi:10.3390/ani12131620_

Round 1
Reviewer 1 Report
The overarching goal of the study was to better understand Salmonella dynamics in slaughter-ready pig cecal material to assess possible public health related outcomes. Furthermore, the study sought to determine the effect of formic acid exposure through drinking water on the prevalence of cecal Salmonella in pigs at the slaughterhouse. The initial purpose of this research was to assess the prevalence of Salmonella and their AMR characteristics in the cecal contents of pigs after slaughter. To meet the first purpose, a total of 25 farms were chosen, and cecal contents of pigs (50 pigs from each farm?) were collected to assess Salmonella prevalence and serotype distribution. The second purpose of this study was to determine the effect of esterified formic acid on Salmonella prevalence in slaughter-ready pig cecal content when formic acid was administered to the pigs via drinking water at the slaughter. For this purpose, eight farms were chosen, and once the pigs were sent to slaughter, 200 pigs were selected from each farm and 40 of those were assigned to the treatment group (where formic acid-containing water made available for the pigs during their wait time until slaughter), while the remaining 160 were kept as the control group, and each group was kept in a separate pre - slaughter pen. This study's focus, in my opinion, is important for improving pork safety. However, the study's design, methods, and analysis all have severe flaws that must be solved before this work is seriously considered for publication.
The manuscript lacked scientific terminology and had serious grammatical and structural language issues. For me, it was confusing, and I believe the whole manuscript needs to be well-structured, organized, edited and evaluated for the use of scientific terminologies and language. Please check below for my main comments and concerns.
The abstract was quite dull and did not adequately describe the relevance of the study, its objectives, results, and conclusions. After addressing some other major flaws (see below), the abstract and research summary must be reconstructed. The authors measured the Salmonella prevalence rather than the shedding. Shedding is considered when the quantitative characteristics of the microorganism are addressed. Authors must use appropriate terminologies while reporting their findings.
There was not much literature background provided in the introduction, which needs to be improved. The introduction should provide scientific evidence as to why this study was conducted and what the major question addressed by this question is. Authors should carefully review the literature and provide more information about the scientific background of formic acid use on food producing animals, as well as safety-related information for both humans and animals (for example, see https://efsa.onlinelibrary.wiley.com/doi/pdf/10.2903/j.efsa.2014.3827 ). Also, since the acid was given to the pigs several hours before slaughter, the authors must justify that the dose of acid given to them was within the EFSA limits and that it does not pose any public health risk. There are other studies that evaluate formic acid on pig microbiota, including Salmonella, that were not mentioned or discussed in the manuscript. This review, for example, addresses some of them https://doi.org/10.3390/ani10050887. To address the evidence-gap and the significance of this study, the authors should conduct a thorough literature review. Again, the introduction section of the manuscript is my least concern right now because my primary concerns are related to the quality of the research and the validity of the results obtained from this study.
The study design was one of the major flaws in the methodology. If this was supposed to be a randomized controlled trial to investigate the "effect of treatment on Salmonella population" the sample size would have been calculated along with the power. The potential biases and confounders (e.g., previous antibiotic exposures, animal weights, uncertain exposure levels, source of the water) that were neither controlled prior to the study, nor considered during the analysis, or discussed are my main concern.
This is not a herd but individual level study focusing on individual pigs clustered in each farms. I see authors use unit, farm, trial, herd for the same concept. This is very confusing. Overall, whole manuscript needs to be checked for consistency of any terms that were referred in multiple sections.
108-115 the link for the questionnaire is not working. I was not able to see the questions. In fact, the survey in the manuscript was irrelevant and made the paper more difficult to understand in terms of the goals. This survey section, in my opinion, needs to be clarified, integrated, and structured, or it should be removed entirely. I'm not sure why Figure 1 was included when the reader has no idea what the questionnaire was about in terms of internal/external biosecurity and how this relates to Salmonella. Furthermore, providing Farm ID and total biosecurity measures makes no sense unless they are integrated with the study's goal, findings, and discussion. Internal and external measures being combined into one does not provide any information. Overall, this survey was completely unrelated to the research.
Again, There was no sample size calculation addressed, the sampling methodology was not well-described or easy to understand.
127 – please provide citation.
128-133 – The use of letters in parenthesis needs to be revised. It is very unclear. Also, I see some information that was provided in the manuscript such as the one on the Line 30 “cefotaxime -5%-, ceftazidime -3.8%- and null for colistin or carbapenems” did not make sense to me. What is “cefotaxime -5%-“? or “null for colistin”? Authors should clarify this or investigate time on the literature and find appropriate way to communicate their research with readers. This is just one example; throughout the manuscript, I see such ambiguous definitions and misuse of terminologies that I am unwilling to address at this stage.
138- needs reference
142- no citation needed for MDR already defined.
144-156 There are also significant flaws in the exposure measurement. Is the treatment dose chosen and potentially administered to these pigs safe for both pigs and humans? This was never mentioned anywhere. What method was used to collect cecal content? The authors should explain how they came to believe that the pigs in the treatment group were drinking the treated water (which was provided in four troughs in the lairage). The authors planned to deliver this water in a controlled manner with a predetermined amount and dose. Was the acid in the water soluble? What is the total amount of acid they are expected to receive per kilogram of pig? Assuming a "dose of exposure" per pig by measuring the leftover water and taking the mean of the water volume that was missing in the troughs is not helpful. It is not a scientific method of determining the level of exposure. This is one of my main concerns about this research. How do the authors know the pigs drank the water that was provided in the pens? What about differences of the body weights of pigs? Also, what is the source of water? Was the content of the water the same each time?
117-122 Instead of providing their Salmonella isolation methodology, they provided a reference that was an ISO standard number. They should detail how they isolate Salmonella and serotype it in their methodology. This is an important part of the manuscript that must be included. I'm also confused as to how they performed their serotyping. It sounds like they performed PCR for detection of S. Typhimurium and Monophasic S. Typhimurium. They simply stated that they used the method described "elsewhere" and provided references. Authors should provide details and clarify how they serotyped the isolated that was recovered. They also used a very ambiguous definition to refer to the types of samples they collected. The sample used should be consistent throughout the manuscript. The serotype names should not be abbreviated. ST is also referred as sequence type which is very confusing for readers. Please provide the serotype names. What is the serotype "other" and how was it typed? As a reader, I was unable to comprehend their serotyping strategy and methodology, as well as the related outcomes.
162-168 this is a part that needs to go to the introduction and discussion not for the methodology
174-177 I am lost here authors already mentioned about Salmonella isolation, I don’t see a reason why this was repeated here.
178-209 The purpose of the survey was to learn about the biosecurity levels of these farms, not to estimate the prevalence of Salmonella. I'm not sure what "percentage of the average resistant" means. Instead, authors should concentrate on whether each isolate is MDR or the number of resistances observed in isolates and their relationship with farms, as well as the frequency and prevalence of resistance of interest. Furthermore, instead of 18, they have 17 antibiotics (line 128-133 and related figure).
Line 187 The formula provided and its contribution to the paper is not clear.
Line 193 Was the Fisher exact test, chi-square test, or logistic regression used? I'm completely lost in this section. Also, what statistical analysis software was used?
Line 195-203 They used multi-level logistic regression rather than multivariable regression. In this data, there are clusters such as farms and seasons that were addressed as random-effect variables. However, the results of their work were misinterpreted (Table 3). They must account for all fixed and random effects and calculate marginal estimates, rather than focusing solely on the CI values calculated for each independent variable included in the final model (either fixed or random effect). Authors also refer to regression models as "global analysis," which is not a proper way to refer to logistic regression in a research paper. Was the model fit evaluated by the authors? What parameters did they come up with? Additionally, rather than assigning only one treatment and control group per farm, authors could consider having triplicates or duplicates within each "trial” to increase the validity.
Line 201 how were the cut off values for time and water consumption determined? What is the biological reasoning behind these values?
Table-2 I am concerned about the authors' background information on the use of the logistic regression model and the interpretation of the results. The logistic regression assumptions related to data were not evaluated prior to data analysis, and the model-fit was not assessed after the model-building strategy, or model parameters were not provided. Instead of considering the full model with fixed and selected random effects, odds ratio interpretations were made at the individual variable level. It's also clear that the data on water consumption was only collected from pigs who were treated, not the entire population. What evidence does the author have that the "observed" difference was due to formic acid exposure? These are legitimate concerns. I'm not sure what "attributable fraction" means in line 205. Now I'm completely lost. Once these major issues have been addressed, I am willing to provide more insight and comments.
Author Response
Please find response attached

Reviewer 2 Report
Dear Mainar Jaime,
I have a few suggestions for the further improvement of your research article entitled "Salmonella shedding in slaughter pigs and the use of organic 2 acids in the drinking water as a potential abattoir-based 3 mitigation measure".
line #15 Almost a third of them (27.3%) shed Salmonella. Change it to Almost one third of them (27.3%) shed Salmonella.
Line #21 "consisting in the addition of 10 kg of formic acid" change it to consisting in the addition of 10 kg formic acid
Line #30 "e. g. cefotaxime -5%-, ceftazidime -3.8%-" change it to "e. g. cefotaxime 5%, ceftazidime 3.8%"
Line #35 "(60.7% in the control group -CG- vs. 44.3% in 35 the treatment group -TG-; P<0.01)" change it to "(60.7% in the control group (CG) vs. 44.3% in 35 the treatment group (TG); P<0.01)"
Line #46 Use European Union when used first time followed by abbreviation (EU).
Line #46 "In 2020, the three more reported Salmonella serovars in human cases in the EU were S. Enteritidis, S. Typhimurium (ST), and the monophasic variant of S. Typhimurium (mST), the latter two being significantly related to pig sources [1]. kindly rewrite this sentence as it is long andconfusing.
Line #50 "and most that did" should be "and those who tried
Line #52 "Salmonella is widely spread among pigs [3]." should be "Salmonella is wide spread among pigs [3]".
Line # 158 "and as indicated in 1.2.". What does 1.2 means??
line #310 "but no one have assessed" should be "but no one has assessed"
Line # 315 "almost a third" should be "almost one third"
Line # 441 "This drop may seem small," should be written as"This drop may seem smaller,"
Line 454 Conclusion section requires complete rephrasing of sentences in order to improve their english and make them understandable.
According to turnitin report the draft has 22% plagiarism and it will be highly appreciable if you may further reduce it to less than 15%.
Author Response
We thank the reviewer for his/her comments
We have considered all of them and modifications have been performed accordingly.
Thanks.